# The clinical value of peripheral immune cell counts in pancreatic cancer

Osama Abu-Shawer[1], Mohammad Abu-Shawer[2], Abdullah Shurman[3], Ali Lattouf[3], Ayman Haimour[4], Omar Hamdan[3], Razan Mansour[3], Tamer Altamimi[3], Maysa Al-Hussaini [5] *

1 Harvard Medical School, Boston, MA, United States of America, 2 Istiklal Hospital, Amman, Jordan, 3 University of Jordan School of Medicine, Amman, Jordan, 4 Department of Biochemistry and Microbiology, University of Victoria, Victoria, Canada, 5 Department of Pathology and Laboratory Medicine, King Hussein Cancer Centre, Amman, Jordan

* mhussaini@khcc.jo

**Data Availability Statement:** All relevant data are within the paper and its Supporting Information files.

**Funding:** The authors received no specific funding for this work.

## Abstract

### Background

Elevated neutrophil-lymphocyte ratio (NLR) is linked to poor overall survival (OS) in pancreatic cancer. We aim to investigate the association of the various hematologic markers, in particular NLR among others, with distant metastases, a common feature in pancreatic cancer.

### Methods

Clinical data from 355 pancreatic cancer patients managed at King Hussein Cancer Center (Amman-Jordan) have been reviewed. We examined the relationship between absolute neutrophil count (ANC), absolute lymphocyte count (ALC), absolute eosinophilic count (AEC), absolute monocytic count (AMC), NLR, monocyte to lymphocyte ratio (MLR) and platelet to lymphocyte ratio (PLR) with the presence of baseline distant metastases and OS. Receiver Operating Characteristic (ROC) curve analysis was plotted to identify the NLR optimum cutoff value indicative of its association with distant metastases.

### Results

On univariate and multivariate analyses patients whom on presentation had high NLR (≥3.3) showed more baseline distant metastases compared to patients with low NLR (<3.3), (p-value: <0.0001 and <0.0001, respectively). Patients with high baseline ANC (≥5500/μL), AMC (≥600/μL), MLR (≥0.3) had more baseline distant metastases in comparison to patients with lower values (p-value: 0.02, 0.001, and <0.0001, respectively). High ANC, NLR, MLR, and PLR and low ALC were associated with poorer OS, (p-value: <0.0001, <0.0001, <0.0001, 0.04, and 0.01, respectively).

### Conclusion

This study presents additional evidence of the association of some of the hematologic markers; in particular ANC, NLR, AMC, and MLR, with baseline distant metastases and poor

**Competing interests:** The authors have declared that no competing interests exist.

outcome in pancreatic cancer. Whether these immune phenomena can help in identifying patients at higher risk for the subsequent development of distant metastases is unknown.

## Introduction

Pancreatic cancer is the most fatal malignancy worldwide with approximately 500,000 deaths each year [1]. More than half of patients had distant metastases at the time of presentation. Despite recent progress in the management of advanced-stage pancreatic cancer, the ultimate prognosis is still dismal [2].

Various studies have associated Inflammation to carcinogenesis, metastases and survival outcomes in cancer. [1, 2] Nonetheless, how inflammation is linked to metastasis remains elusive. Hematologic parameters, in particular neutrophil-lymphocyte ratio (NLR), were repeatedly shown to negatively impact survival outcome in various tumors including pancreatic cancer [3–8]. Additionally, studies have related high NLR to baseline presence and subsequent development of distant metastases in various tumors including non-small cell lung cancer, gastric cancer, and colorectal cancer among others [9–11]. Theories were proposed in an attempt to explain the relationship between inflammation and metastasis. A high circulating neutrophil count, which would result in a high NLR, might promote tumor cells' metastasis via releasing various proteases and specific growth factors like vascular endothelial growth factor (VEGF) [13]. Moreover, circulating lymphocytes induce cytotoxic cell death, and therefore are actively involved in the elimination of tumor cells, in addition to delaying tumor cell migration, thus explaining the relationship between low ALC and poor outcomes [14, 15].

In this retrospective study, we aim to investigate the relationship between the various peripheral hematologic cells, in particular the different types of inflammatory cells with distant metastases in pancreatic cancer. Our main objectives are to answer the following questions; Do pancreatic cancer patients with high NLR have a high incidence of distant metastases? Would they benefit from more frequent surveillance investigation including imaging? And finally, would they potentially benefit from prophylactic strategies?

## Methods

### Initial step in data collection from the electronic medical records

This is a retrospective chart review study approved by the Institutional Review Board (IRB) at King Hussein Cancer Center (KHCC). Once the IRB approval was granted, a request was made to the Center's Cancer Registry to obtain the full details of patients diagnosed and managed i.e. full abstract, with pancreatic carcinoma from July 2006- December 2018 inclusive. The Electronic Medical Records (EMRs) at KHCC were then searched for particular hematologic parameters. The IRB policy that governs the ethical approval for studies on pre-existing i.e. data is already documented in the EMR and/ or Center's Cancer Registry at the time of data collection does not mandate patient's informed consent in the absence of direct interaction/ intervention. Positron emission tomography (PET), computed tomography (CT), and magnetic resonance imaging (MRI) scans were reviewed to detect the presence of baseline distant metastases. Clinical data including age, gender, location of the primary tumor, and sites of baseline distant metastases are summarized in Table 1. We excluded all patients who received only part of treatment at our center and patients who were on steroids before obtaining a complete blood count (CBC). Using steroids was expected to confound the results of the study as it leads to leukocytosis and specifically neutrophilia [16, 17].

**Table 1. Characteristics of pancreatic cancer patients.**

| Patients features | No. of patients (%) |
|---|---|
| Age > 60 | 167 (47%) |
| Age< = 60 | 188 (53%) |
| Gender | |
| Male | 210 (59%) |
| Female | 145 (41%) |
| Location | |
| Head of pancreas | 184 (52%) |
| Others (body, tail. . .etc.) | 110 (31%) |
| Unknown | 61 (17%) |
| Baseline Distant Metastases | |
| Present | 206 (58%) |
| Absent | 149 (42%) |
| Clinical TNM Stage | |
| I & II | 43 (12%) |
| III | 106 (30%) |
| IV | 206 (58%) |
| Hematologic Parameters: | Median, (Mean) |
| ANC | 5500 (6550) |
| ALC | 1680 (1900) |
| AEC | 140 (190) |
| AMC | 600 (670) |
| NLR | 3.3 (4.3) |
| MLR | 0.3 (0.4) |
| PLR | 0.15 (0.2) |

Complete blood count (CBC) with the differential white cell count was collected before the initiation of any cancer-specific treatment (systemic treatment or radiation). The pre-treatment baseline NLR, MLR and PLR were calculated using these formulas; NLR = ANC/ALC, MLR = AMC/ALC and PLR = Platelet Count/ALC.

## Determination of the NLR cut-off value and relation to other variables

The Receiver Operating Characteristic (ROC) curve was operated to determine the best NLR cut-off value, for assessing its association with the presence of baseline distant metastases, matching the most extreme joint sensitivity and specificity. The association between NLR, age, gender and the anatomic location of the primary tumor within the pancreas with the presence of baseline distant metastases was tested. Univariate and multivariate logistic regression analyses were used to examine the association between the various variables and the presence of baseline distant metastases. A p-value of $\leq 0.05$ was determined as the cut off value for significance association.

## The association between baseline hematological markers and distant metastasis

Our analysis proceeded stepwise. In the first phase, we examined the association between baseline NLR with the presence of distant metastases. In the second phase, we examined the association between other hematologic parameters including ANC, ALC, AEC, AMC, MLR and PLR with the baseline presence of distant metastases.

**Table 2. The association between hematologic indices with the presence of baseline distant metastases.**

| | Baseline distant metastases | | | | |
|---|---|---|---|---|---|
| | Present | Absent | p- value | OR | 95% CI |
| Baseline ANC≥5500 | 113 (64%) | 63 (36%) | 0.024 | 1.65 | (1.0–2.5) |
| Baseline ANC<5500 | 92 (52%) | 85 (48%) | | | |
| Baseline ALC≥1680 | 99 (57%) | 75 (43%) | 0.65 | 0.9 | (0.6–1.4) |
| Baseline ALC<1680 | 106 (59%) | 73 (41%) | | | |
| Baseline AMC≥ 600 | 117 (66%) | 59 (34%) | 0.001 | 2.0 | (1.3–3.0) |
| Baseline AMC< 600 | 88 (50%) | 89 (50%) | | | |
| Baseline AEC≥ 143 | 83 (62%) | 51 (38%) | 0.38 | 1.2 | (0.7–2.0) |
| Baseline AEC< 143 | 76(57%) | 58 (43%) | | | |
| Baseline NLR≥3.3 | 126 (69%) | 56 (31%) | < .0001 | 2.6 | (1.7–4.0) |
| Baseline NLR<3.3 | 79 (46%) | 92 (54%) | | | |
| Baseline MLR≥0.3 | 130 (68%) | 62 (32%) | < .0001 | 2.4 | (1.5–3.7) |
| Baseline MLR<0.3 | 75 (46%) | 86 (54%) | | | |
| Baseline PLR≥0.15 | 108 (61%) | 68 (39%) | 0.2 | 1.3 | (0.85–2.0) |
| Baseline PLR<0.15 | 97 (55%) | 80 (45%) | | | |

## The association between baseline hematological markers and clinical variables

In the third phase, we examined the association between baseline presence of distant metastases with the clinical variables like age, gender, and location of the primary tumor.

In the fourth phase, we performed a multivariate analysis that included the collected variables (age, gender, and location of the primary tumor) with NLR as a dichotomous variable, to identify correlation with baseline distant metastases.

## The association between baseline hematological markers and overall survival

In the fifth phase, we examined the association between the hematologic parameters including ANC, ALC, AEC, AMC, NLR, MLR & PLR with the OS. In the last phase, we examined the

**Table 3. Univariate and multivariate analyses for the association of different variables with the presence of baseline distant metastases.**

| | Baseline Distant Metastases | | Univariate | | Multivariate | |
|---|---|---|---|---|---|---|
| Variables | Present | Absent | P value | OR (95% CI) | P value | OR (95% CI) |
| Age | | | 0.29 | | 0.02 | |
| Age≤60 | 114 (61%) | 74 (39%) | | 1.2 (0.8–1.9) | | 1.8 (1.0–2.9) |
| Age>60 | 92 (55%) | 75 (45%) | | | | |
| Gender | | | 0.53 | | 0.9 | |
| Male | 119 (57%) | 91 (43%) | | 0.8 (0.5–1.3) | | 0.9 (0.6–1.6) |
| Female | 87 (60%) | 58 (40%) | | | | |
| Location | | | < .0001 | | < .0001 | |
| Head | 85 (46%) | 99 (54%) | | 0.3 (0.19–0.53) | | 0.3 (0.18–0.54) |
| Others | 80 (73%) | 30 (27%) | | | | |
| Unknown | 41 (67%) | 20 (33%) | | | | |
| Baseline NLR | | | < .0001 | | 0.0003 | |
| NLR>3.3 | 126 (69%) | 56 (31%) | | 2.6 (1.7–4.0) | | 2.5 (1.5–4.1) |
| NLR≤3.3 | 79 (46%) | 92 (54%) | | | | |

**Table 4. The association between NLR (categorical variable) and the presence of baseline distant metastases.**

| NLR | Distant Mets Present | Distant Mets Absent | Odds Ratio | 95% Confidence Intervals |
|---|---|---|---|---|
| NLR (<2.0) | 39 (50%) | 39 (50%) | 1.0 | N/A |
| NLR (2.0–3.2) | 45 (45%) | 54 (55%) | 0.833 | (0.460–1.510) |
| NLR (3.2–5.0) | 60 (70%) | 26 (30%) | 2.307 | (1.217–4.373) |
| NLR (>5.0) | 61 (68%) | 29 (22%) | 2.103 | (1.124–3.935) |

association between NLR as a categorical variable with both the presence of baseline distant metastases and the OS.

Descriptive analysis of the studied patients' information was performed. Categorical data, such as age group, gender and other factors were presented as counts and percentages. The mean, standard deviation and range were calculated for the continuous data including age, ANC and other factors. In general, differences in proportions were tested with $\chi^2$ test or Fisher's exact test, and differences in continuous variables were tested with Student's t-test or a non-parametric test (Wilcoxon Rank Test) depending on the assumptions required for each test. Multivariate analysis was done for gender, age, site (pancreas head vs others), NLR cutoff (NLR>3.286 vs NLR< = 3.286) with probability modeled on (stage = 'IV'), using the logistic regression model. Kaplan-Meier method was used to estimate OS curves, and log-rank test was used to compare patients' survival times between factors groups. The OS time was calculated from the diagnosis date to death from any cause. Survival was expressed as median with a 95% confidence interval (CI). Multivariate analysis was done by using Cox proportional hazards regression model. Receiver operating characteristic curve (ROC curve) was performed for NLR ANC, AMC, and MLR with stage and survival. A significance criterion of $p< = 0.05$ was used in the analysis. All analyses were performed using SAS version 9.4 (SAS Institute Inc, Cary, NC).

## Results

The clinical characteristics of 355 pancreatic cancer patients are summarized in Table 1. More than half of the patients were males; (59% versus 41.0%, with a male to female ratio of 1.4:1).

**Table 5. The results of the survival analysis.**

| | Median of OS | Number of patients | p-value | HR | 95% CI |
|---|---|---|---|---|---|
| ANCv5500 | 5 months | 176 (50%) | < .0001 | 1.5 | (1.2–1.9) |
| ANC < 5500 | 10 months | 177 (50%) | | | |
| ALC≥ 1680 | 9 months | 174 (49%) | 0.01 | 0.7 | (0.6–0.9) |
| ALC < 1680 | 5 months | 179 (51%) | | | |
| AEC≥143 | 8 months | 134 (50%) | 0.54 | 0.9 | (0.7–1.2) |
| AEC < 143 | 8 months | 134 (50%) | | | |
| AMC≥ 600 | 6 months | 176 (50%) | 0.2 | 1.3 | (0.9–1.4) |
| AMC < 600 | 8 months | 177 (50%) | | | |
| NLR≥ 3.3 | 4 months | 176 (50%) | < .0001 | 1.9 | (1.5–2.5) |
| NLR < 3.3 | 11 months | 177 (50%) | | | |
| MLR≥ 0.3 | 5 months | 192 (54%) | < .0001 | 1.5 | (1.2–1.9) |
| MLR < 0.3 | 9 months | 161 (46%) | | | |
| PLR≥ 0.15 | 5 months | 176 (50%) | 0.04 | 1.2 | (1.0–1.5) |
| PLR < 0.15 | 9 months | 177 (50%) | | | |

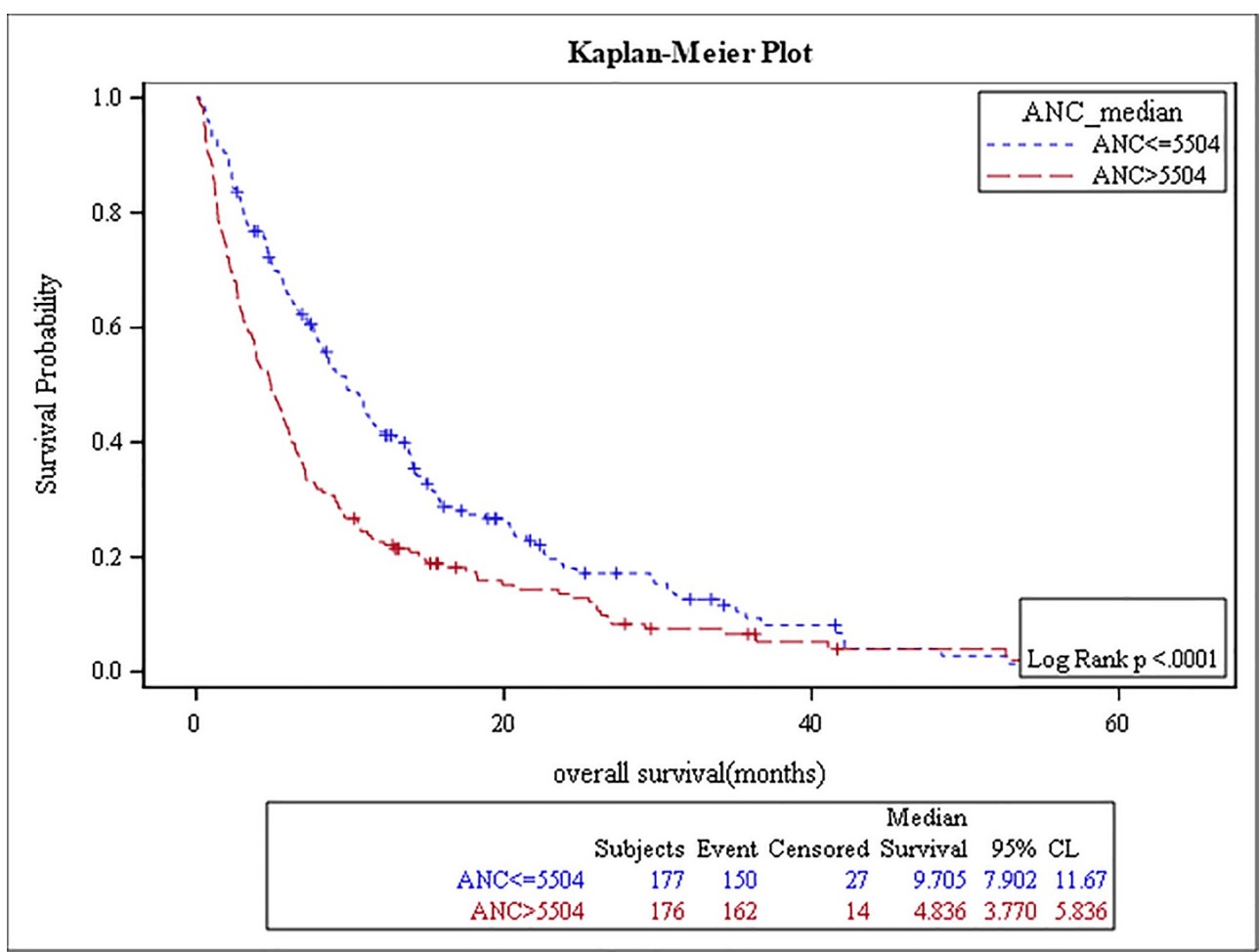

**Fig 1. Kaplan Meier curve for overall survival for patients with ANC≥ 5500.**

The median age at diagnosis was 60 years. The median OS for all patients was 6.6 months. More than half of the patients had distant metastases at time of presentation (n = 206, 58%). The mean of baseline NLR was 4.3, and the median was 3.3. The cutoff value of NLR for its association with baseline distant metastases was determined to be 3.3 using the ROC curve, where the area under the curve (AUC) had a value of 0.609 (S1 Fig).

The relationship between the peripheral count of various immune cells, their ratios and the baseline presence of distant metastases are shown in Table 2. Patients with elevated baseline NLR (≥3.3) were more likely to have distant metastases at time of presentation in comparison to patients with low baseline NLR (<3.3), (p-value <0.0001, Odds Ratio (OR): 1.7, CI: 2.6–4.0). High baseline ANC (≥5500/μL), high AMC (≥600 /μL), and high MLR (≥0.3) were associated with baseline distant metastases (p-value: 0.02, 0.001, and <0.0001 respectively). ALC, AEC, and PLR were not associated with the presence of baseline distant metastases in pancreatic cancer (p-value: 0.65, 0.38 and 0.21, respectively).

Table 3 shows the univariate and multivariate analyses assessing the association of NLR and clinical variables with baseline distant metastases. The location of the primary tumor was significantly associated with the baseline presence of distant metastases; patients with the primary tumor located in the head of the pancreas had less baseline distant metastases compared to patients with the primary tumor located in other sites (body, tail, and overlapping), (p-value:

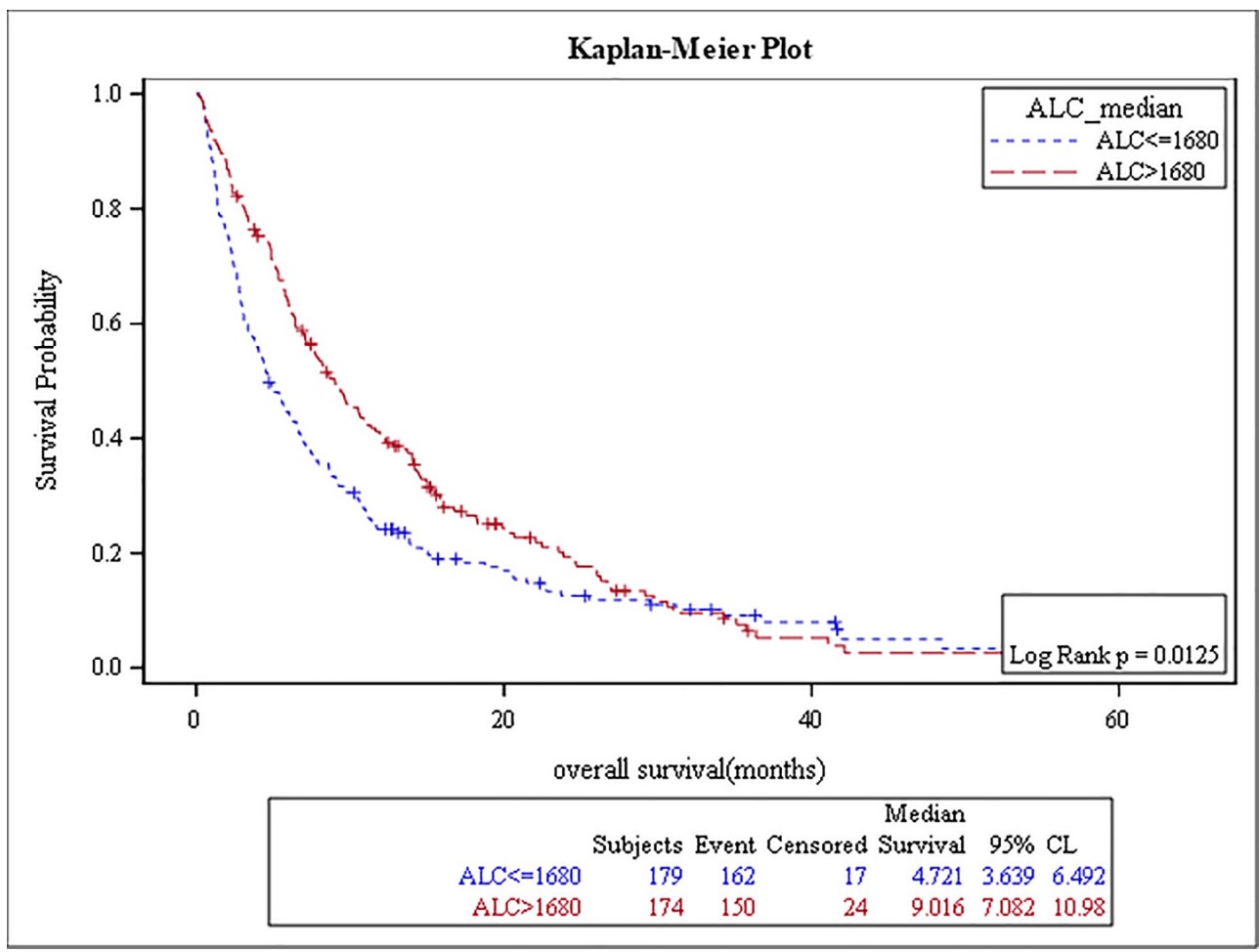

**Fig 2. Kaplan Meier curve for overall survival for patients with ALC≥ 1680.**

<0.0001). Age and gender were not associated with baseline presence of distant metastases in the univariate analysis (p-value: 0.29 and 0.53, respectively). In spite of that, we involved both factors in the multivariate analysis because of their clinical relevance. In the multivariate analysis, age, location of the primary tumor, and NLR, but not gender were significantly associated with the baseline presence of distant metastases (p-value: 0.02, <0.0001, 0.0003, and 0.9, respectively). Furthermore, the association with baseline metastasis remained significant even when NLR was evaluated as a categorical variable. (Table 4)

The results of the survival analysis are summarized in Table 5. High baseline ANC, NLR, MLR, PLR, and low ALC were associated with poor OS, (p-value: <0.0001, <0.0001, <0.0001, 0.04, and 0.01, respectively) (Figs 1–5). The ROC curve was utilized to determine if NLR, ANC, ALC, MLR, or PLR is a better predictor of overall survival. The curve showed that NLR has the highest AUC followed by ANC, PLR, MLR and then ALC which indicates that NLR is a better predictor of overall survival in comparison to these tested variables (S2 Fig). High NLR was also associated with poor OS as a categorical variable (Table 6 & Fig 6). Baseline AEC and AMC were not associated with the OS, (p-value: 0.54 and 0.23, respectively).

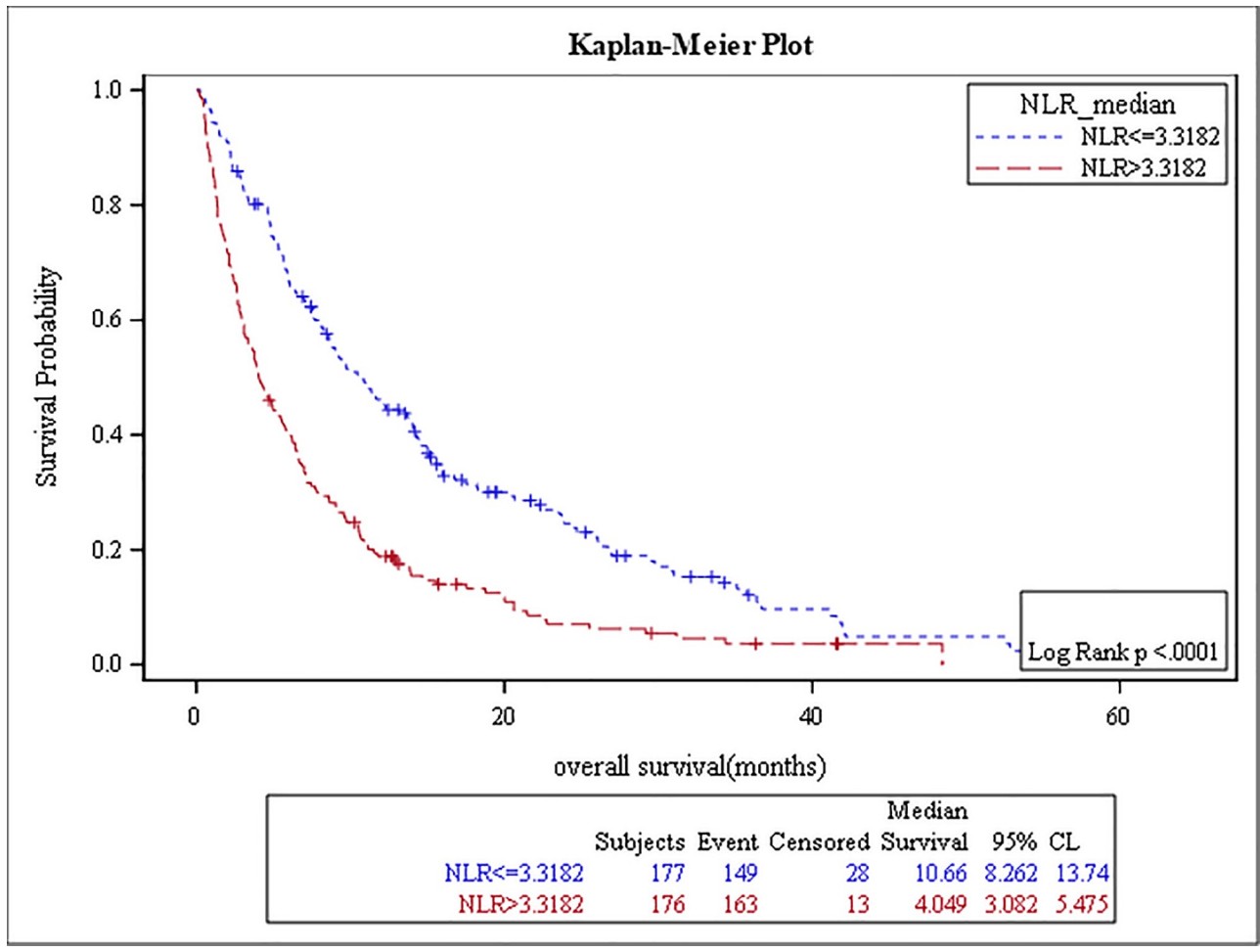

**Fig 3. Kaplan Meier curve for overall survival for patients with NLR≥3.3.**

## Discussion

In this study we assessed the association of NLR, ANC, ALC, AEC, AMC, MLR and PLR with the presence of baseline distant metastases and OS in pancreatic cancer patients, as well as the potential predictive value of NLR for detecting distant metastases. An elevated baseline NLR ($\geq$ 3.3) was an independent factor associated with the baseline presence of distant metastases in pancreatic cancer, as depicted in both the univariate and multivariate analyses, after adjusting for important covariates (p = < .0001, and p = 0.0003, respectively) (Table 2). Patients with elevated baseline ANC ($\geq$5500/μL), AMC ($\geq$600/μL), MLR ($\geq$0.3) had more distant metastases in comparison to patients with low baseline ANC (<5500/μL), AMC (<600/μL), MLR (<0.3) (p-value: 0.02, 0.001 and <0.0001, respectively).

Several studies have investigated the prognostic value of these inflammatory markers in pancreatic cancer, in which NLR, PLR, MLR were reported to be associated with poor OS. The cutoff value of NLR reported in these studies varies widely; ranging from 2.3 to 5[12]. In our cohort, the cutoff value of NLR using the ROC curve is 3.3. To our knowledge, this appears to be the first study exploring the association between these different inflammatory markers with the presence of baseline distant metastatic lesions in pancreatic cancer.

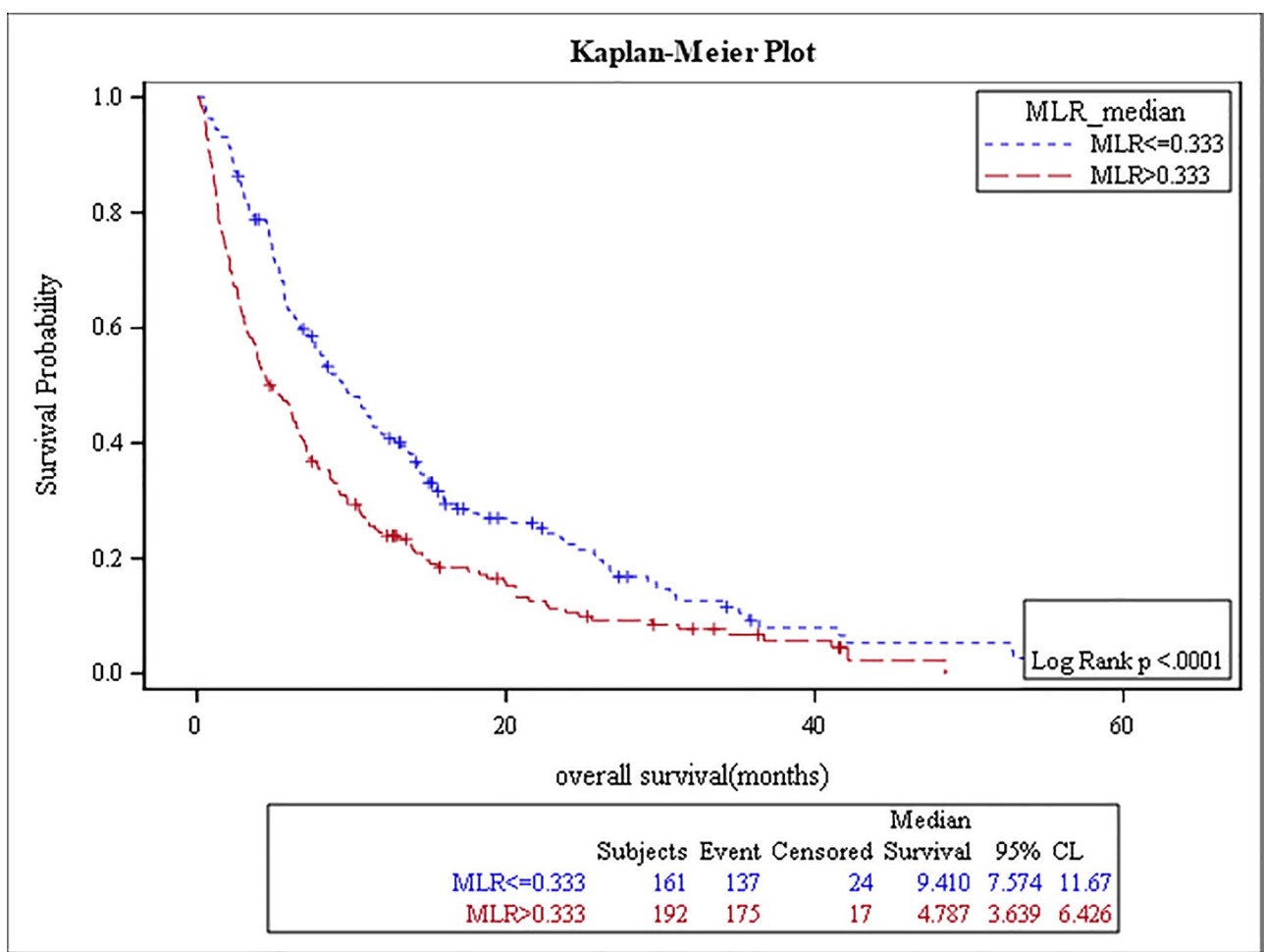

**Fig 4. Kaplan Meier curve for overall survival for patients with MLR≥0.3.**

Using a simple test such as a complete blood count (CBC), to foresee the likelihood of the development distant metastases might have a direct impact on the management of pancreatic cancer patients through early detection of distant metastatic lesions. The presence of high NLR in pancreatic cancer patients' CBC tests could stimulate surveillance for distant metastases through more frequent imaging, and consideration for prophylactic measures.

In patients with different types of malignant tumors, NLR correlates with the extent of therapeutic success in response to chemotherapy and immunotherapy [13–17]. Moreover, other blood-based markers i.e. circulating tumor cells and tumor-DNA were reported to be associated with survival in pancreatic cancer [8]. This might represent a prospect to link these markers together and study their potential associations.

Treatment plans to limit distant metastases in pancreatic cancer include chemotherapy, radiation, and surgery. Future research can include investigating the association of NLR to the response of distant metastases to the various treatment modalities. A major advancement in the management of advanced-stage pancreatic cancer will be achieved if NLR is confirmed to be a predictive marker of distant metastases' response to available and future novel treatments.

We acknowledge that the retrospective nature of the data, which was collected from a single institution, represents a limitation to this study. Moreover, there was no assessment of the

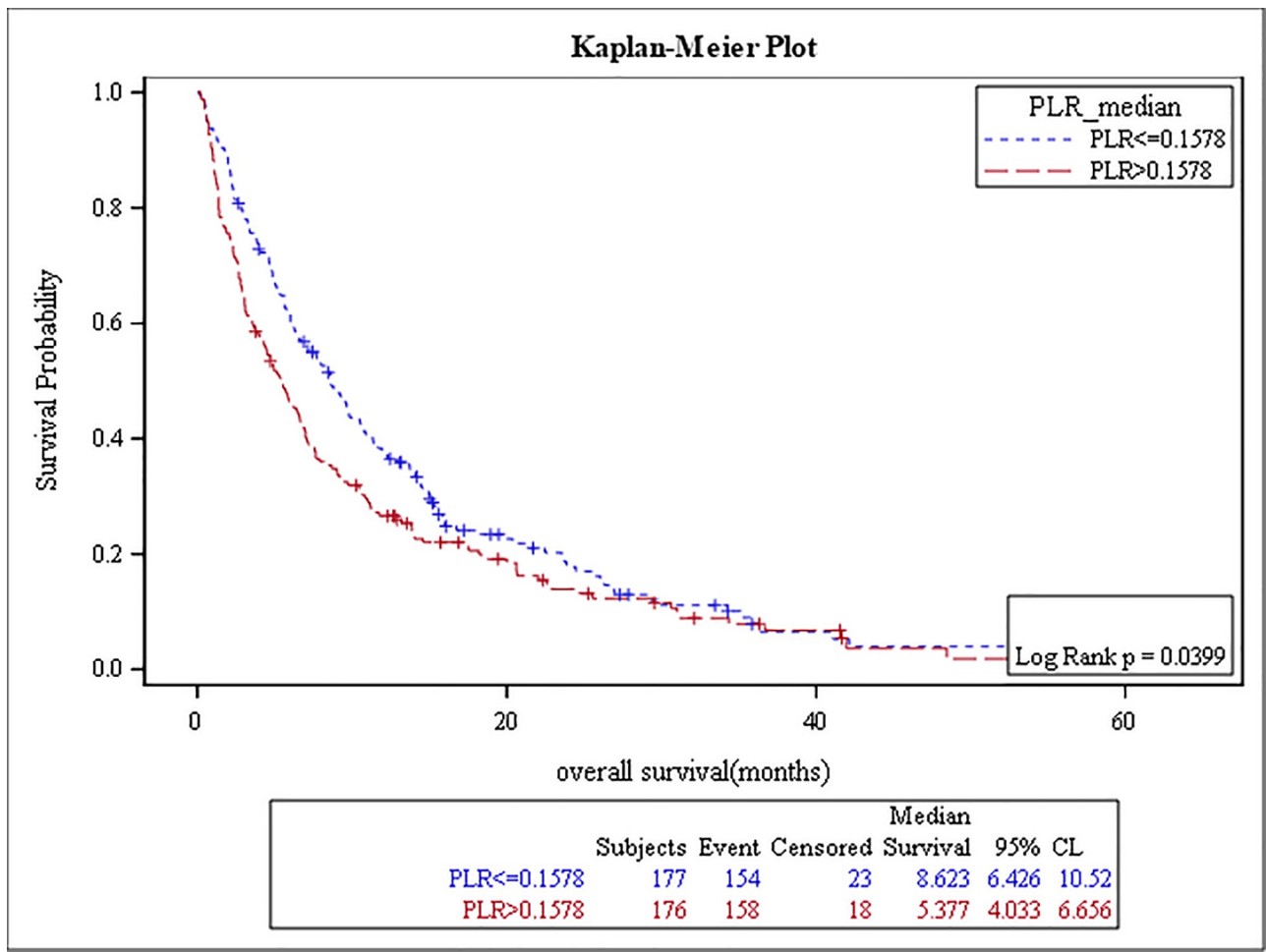

**Fig 5. Kaplan Meier curve for overall survival for patients with PLR≥ 0.15.**

exact treatments administered and of any other factors that might alter NLR, such as infections. Nonetheless, we believe that our study adds more evidence to current literature on the potential usefulness of hematological markers in predicting distant metastases in pancreatic cancer, and which might potentially be useful in assessing the response to various treatment modalities including novel therapies.

## Conclusion

In pancreatic cancer, elevated baseline NLR, ANC, AMC, and MLR appear to be independent predictive factors for the presence of baseline distant metastases. Prospective validation and

**Table 6. The relationship between NLR (categorical variable) and OS.**

| NLR | Overall Survival (mons) | p-value | Hazard Ratio | 95% Confidence Intervals |
|---|---|---|---|---|
| NLR (<2.0) | 9.7 | 0.0001 | 1.0 | N/A |
| NLR (2.0–3.2) | 11.3 | | 0.979 | (0.701–1.368) |
| NLR (3.2–5.0) | 5.2 | | 1.770 | (1.262–2.483) |
| NLR (>5.0) | 3.0 | | 2.164 | (1.548–3.027) |

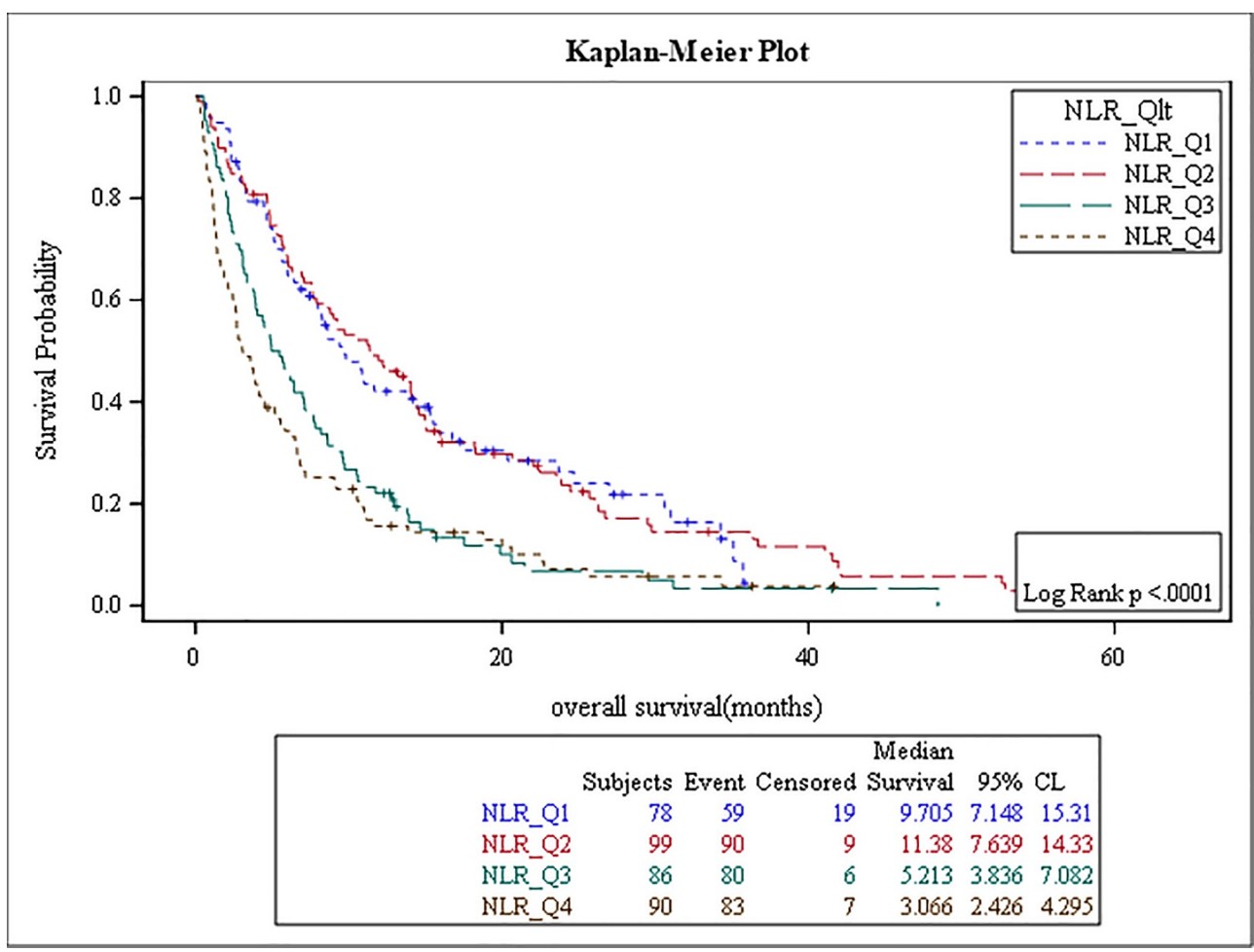

**Fig 6. Kaplan Meier curve for overall survival for patients with NLR as categorical variable.**

how these inflammatory markers can be related to other markers to predict the likelihood of subsequent development of distant metastases warrants further investigation.

## Supporting information

**S1 Data.**
(XLS)

**S1 Fig. Receiver-operating-characteristic (ROC) curve and the area under the curve (AUC) for NLR.**
(TIF)

**S2 Fig. ROC curve of NLR, ANC, ALC, MLR, and PLR for predicting overall survival.**
(TIF)

## Acknowledgments

The authors would like to thank Mrs. Ayat Taqash for her help in statistical analysis.

## Author Contributions

**Conceptualization:** Osama Abu-Shawer, Mohammad Abu-Shawer, Ayman Haimour, Tamer Altamimi.

**Data curation:** Abdullah Shurman, Ali Lattouf, Ayman Haimour, Omar Hamdan, Razan Mansour, Tamer Altamimi.

**Formal analysis:** Osama Abu-Shawer.

**Investigation:** Abdullah Shurman, Ali Lattouf, Omar Hamdan, Tamer Altamimi.

**Methodology:** Osama Abu-Shawer.

**Project administration:** Maysa Al-Hussaini.

**Supervision:** Maysa Al-Hussaini.

**Writing – original draft:** Osama Abu-Shawer, Ayman Haimour, Razan Mansour.

**Writing – review & editing:** Mohammad Abu-Shawer, Ayman Haimour, Razan Mansour, Maysa Al-Hussaini.

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
