## [Decision Letter · Decision Letter 0]

10 Feb 2020

PONE-D-20-00008

The Clinical Value of Peripheral Immune Cell Counts in Pancreatic Cancer

PLOS ONE

Dear Al-Hussaini,

Thank you for submitting your manuscript to PLOS ONE. After careful consideration, we feel that it has merit but does not fully meet PLOS ONE’s publication criteria as it currently stands. Therefore, we invite you to submit a revised version of the manuscript that addresses the points raised during the review process.

We would appreciate receiving your revised manuscript by Mar 26 2020 11:59PM. To enhance the reproducibility of your results, we recommend that if applicable you deposit your laboratory protocols in protocols.io, where a protocol can be assigned its own identifier (DOI) such that it can be cited independently in the future. For instructions see: http://journals.plos.org/plosone/s/submission-guidelines#loc-laboratory-protocols

We look forward to receiving your revised manuscript.

Kind regards,

Aldo Scarpa

Academic Editor

PLOS ONE

Journal Requirements:

2. We noticed minor instances of text overlap with the following previous publication(s), which need to be addressed:

(1) http://jgo.amegroups.com/article/view/26936

The text that needs to be addressed involves the Abstract and Discussion section.

In your revision please ensure you cite all your sources (including your own works), and quote or rephrase any duplicated text outside the methods section. Further consideration is dependent on these concerns being addressed.

3. In the ethics statement in the manuscript and in the online submission form, please provide additional information about the patient records used in your retrospective study, including: a) whether all data were fully anonymized before you accessed them; b) the date range (month and year) during which patients' medical records were accessed; c) the date range (month and year) during which patients whose medical records were selected for this study sought treatment; and d) the source of the medical records analyzed in this work (e.g. hospital, institution or medical center name). If patients provided informed written consent to have data from their medical records used in research, please include this information.

4. At this time, we ask that you please include sub-section titles in your Methods section to aid in the reporting and reading of the manuscript.

"None"

Please provide an amended Funding Statement that declares *all* the funding or sources of support received during this specific study (whether external or internal to your organization) as detailed online in our guide for authors at http://journals.plos.org/plosone/s/submit-now.  Please state what role the funders took in the study.  If any authors received a salary from any of your funders, please state which authors and which funder. If the funders had no role, please state: "The funders had no role in study design, data collection and analysis, decision to publish, or preparation of the manuscript."

6. Thank you for stating the following in your Competing Interests section: 

"None"

7. We note that you have indicated that data from this study are available upon request. PLOS only allows data to be available upon request if there are legal or ethical restrictions on sharing data publicly. For information on unacceptable data access restrictions, please see http://journals.plos.org/plosone/s/data-availability#loc-unacceptable-data-access-restrictions.

Reviewers' comments:

Reviewer's Responses to Questions

**Comments to the Author**

1. Is the manuscript technically sound, and do the data support the conclusions?

Reviewer #1: Yes

Reviewer #2: Partly

2. Has the statistical analysis been performed appropriately and rigorously? 

Reviewer #1: Yes

Reviewer #2: Yes

3. Have the authors made all data underlying the findings in their manuscript fully available?

Reviewer #1: Yes

Reviewer #2: Yes

4. Is the manuscript presented in an intelligible fashion and written in standard English?

Reviewer #1: Yes

Reviewer #2: Yes

5. Review Comments to the Author

Reviewer #1: This manuscript aims at evaluating the potential prognostic role in pancreatic cancer of the absolute count of some inflammatory cell types (neutrophils, lymphocytes, eosinophils and monocytes), as well as of two “absolute” values, such as monocyte-lymphocyte ratio and platelet-lymphocyte ratio.

This paper is well written and has a very good sample size (355 patients); the ROC curves /statistics have been properly used. In my opinion, there are some points to be improved before considering this paper for publication, as follows:

- In the abstract, mini-abstract and in text, the authors should tone down the considerations regarding immunotherapy. For example, they have written: “whether or not these immune phenomena can help in identifying patients who might respond to immunotherapy needs further investigation”. This consideration should be deleted, since does not have much sense. Indeed, immunotherapy in pancreatic cancer does not represent a possible therapeutic strategy in the vast majority of cases (>98%), due to the very rare presence of microsatellite instability and of high-tumor mutational burden (and also the lack of significant utility of PD-L1 as a predictive marker). The importance of this study, in the opinion of this reviewer, is giving further biological insights on the blood parameters of patients with pancreatic cancer and demonstrating that other blood related variables (other than the classical known) may be introduced into the clinical practice due to their potential diagnostic / prognostic value. All the implications and considerations about immunotherapy, thus, should be deleted or toned down.

- For investigating the “prognostic” role of the studied parameters, the authors used the baseline presence of metastasis. It may be of great help if they have also values (timing) on metastatic progression for calculating this “prognostic role” also as the effective metastatic risk.

- In the discussion, the authors should better explain the importance and future perspectives of blood analysis for patients with pancreatic cancer. Indeed, not only inflammatory-cells related values can be investigated, but also other parameters, such as tumor circulating cells and circulating tumor-DNA, may be of great importance. Please discuss more in depth this topic (I suggest this comprehensive reference: PMID: 31405192)

- In the discussion, please comment more in depth the prognostic value of neutrophil-to-lymphocyte ratio for invasive pancreatic cancers associated to IPMN. You may use for this the main reference on this topic, which from the group of Dr. Christopher Wolfgang from the Johns Hopkins University (Gemenetzis G et al.; Ann Surg. 2017; PMID: 27631774). This reference should be used also in the introduction when the authors presented similar moderators in different cancer types.

- Figure 1 and figure 7 should be put as “supplementary figures”. There are 6 figures (Kaplan-Meier curves) that are very important for the reader, but the two figures presenting the “ROC curves” are not so important for the reader. Indeed, they represent a “concept” very important for the methods, but not for the overall comprehension of the paper. Thus, please put them as supplementary (as usually happens).

Reviewer #2: The manuscript by Wang and co-authors titled “The clinical value of peripheral immune cell counts in pancreatic cancer” demonstrated the potential role of the absolute numbers enumeration of myeloid and lymphocytes as a potential biomarker to predict distant metastases. The major concern of this work is the novelty. Several reports have already reported these parameters (NLR, MLR etc) to identify PDAC patients with a poor prognosis induced by metastatic disease: Formica V. et al. Neutrophil/lymphocyte Ratio Helps Select Metastatic Pancreatic Cancer Patients Benefitting From Oxaliplatin, Cancer Biomark; 2016; Ventriglia J. et al. Neutrophil to Lymphocyte Ratio as a Predictor of Poor Prognosis in Metastatic Pancreatic Cancer Patients Treated with Nab-Paclitaxel plus Gemcitabine: A Propensity Score Analysis, Gastroenterology Research and Practice; 2018; Piciucchi M. et al. The Neutrophil/Lymphocyte Ratio at Diagnosis Is Significantly Associated with Survival in Metastatic Pancreatic Cancer Patients, Int J Mol Sci., 2017.

The authors did not comment their results in light to the published data, for example: the identified cutoff of NLR (3.3) is quite different compared to data from literature. Please comment this discrepancy.

More concerns are about patient data. As reported the patient cohort is based on 355 patients. Considering the percentage of absent metastases at baseline from Table 2, the reader can conclude that this experimental group is approximately composed by 178 patients (i.e 89 patients represent 50% of the group). However from Table 1, patients with baseline metastases (Liver+ lung + peritoneal) are listed as 260. Therefore, the analyzed patient cohort result to be composed by 438 patients. Please comment and clarify this aspect.

Minor concern. Some conclusions are overstated:

Abstract. “…these immune phenomena can help in identifying patients who might respond to immunotherapy”. In my knowledge there are not clinical conventional immune-based approaches for PDAC treatment.

Finally, the graphic quality is very poor.

---

## [Author Response · Author response to Decision Letter 0]

23 Mar 2020

Editorial Formatting comments: 

All the suggestions regarding editorial formatting have been incorporated. All paragraphs were re-written to minimize any potential overlap with our previously published work. 

Response to Reviewer 1: 

Thank you for your review of our paper. We have answered each of your points below. 

1. In the abstract, mini-abstract and in text, the authors should tone down the considerations regarding immunotherapy. For example, they have written: “whether or not these immune phenomena can help in identifying patients who might respond to immunotherapy needs further investigation”. This consideration should be deleted, since does not have much sense. Indeed, immunotherapy in pancreatic cancer does not represent a possible therapeutic strategy in the vast majority of cases (>98%), due to the very rare presence of microsatellite instability and of high-tumor mutational burden (and also the lack of significant utility of PD-L1 as a predictive marker). The importance of this study, in the opinion of this reviewer, is giving further biological insights on the blood parameters of patients with pancreatic cancer and demonstrating that other blood related variables (other than the classical known) may be introduced into the clinical practice due to their potential diagnostic / prognostic value. All the implications and considerations about immunotherapy, thus, should be deleted or toned down.

Response: Done. We have removed all the sentences that addressed the use of immunotherapy in pancreatic cancer.

2. For investigating the “prognostic” role of the studied parameters, the authors used the baseline presence of metastasis. It may be of great help if they have also values (timing) on metastatic progression for calculating this “prognostic role” also as the effective metastatic risk.

Response: We agree that investigating the predictive value of the baseline immune cell counts for the subsequent development of distant metastases would be of a great help, but since a small percentage of our cohort had no baseline distant metastases and no available data regarding the subsequent development of distant metastases in this subset of patients, this was not feasible.

3. In the discussion, the authors should better explain the importance and future perspectives of blood analysis for patients with pancreatic cancer. Indeed, not only inflammatory-cells related values can be investigated, but also other parameters, such as tumor circulating cells and circulating tumor-DNA, may be of great importance. Please discuss more in depth this topic (I suggest this comprehensive reference: PMID: 31405192)

Response: Done. We touched base with other potential markers of distant metastases in the discussion and cited the suggested reference as well.

4. In the discussion, please comment more in depth on the prognostic value of neutrophil-to-lymphocyte ratio for invasive pancreatic cancers associated to IPMN. You may use for this the main reference on this topic, which is from the group of Dr. Christopher Wolfgang from the Johns Hopkins University (Gemenetzis G et al.; Ann Surg. 2017; PMID: 27631774). This reference should be used also in the introduction when the authors presented similar moderators in different cancer types.

Response: Done. We have cited this reference in the introduction. Our aim in this study was to explore the predictive value of these inflammatory-based markers for the presence of distant metastases in pancreatic cancer. We can’t discuss other predictive values of these markers in this paper in depth as that would be irrelevant topics. 

5. Figure 1 and figure 7 should be put as “supplementary figures”. There are 6 figures (Kaplan-Meier curves) that are very important for the reader, but the two figures presenting the “ROC curves” are not so important for the reader. Indeed, they represent a “concept” very important for the methods, but not for the overall comprehension of the paper. Thus, please put them as supplementary (as usually happens).

 Response: Done. We made the ROC curves supplementary figures.

Response to Reviewer 2: 

Thank you for your review of our paper. We have answered each of your points below. 

1. The manuscript by Wang and co-authors titled “The clinical value of peripheral immune cell counts in pancreatic cancer” demonstrated the potential role of the absolute numbers enumeration of myeloid and lymphocytes as a potential biomarker to predict distant metastases. The major concern of this work is the novelty. Several reports have already reported these parameters (NLR, MLR etc) to identify PDAC patients with a poor prognosis induced by metastatic disease: Formica V. et al. Neutrophil/lymphocyte Ratio Helps Select Metastatic Pancreatic Cancer Patients Benefitting From Oxaliplatin, Cancer Biomark; 2016; Ventriglia J. et al. Neutrophil to Lymphocyte Ratio as a Predictor of Poor Prognosis in Metastatic Pancreatic Cancer Patients Treated with Nab-Paclitaxel plus Gemcitabine: A Propensity Score Analysis, Gastroenterology Research and Practice; 2018; Piciucchi M. et al. The Neutrophil/Lymphocyte Ratio at Diagnosis Is Significantly Associated with Survival in Metastatic Pancreatic Cancer Patients, Int J Mol Sci., 2017.

Response: We agree that there are articles in the literature addressing the prognostic utility of NLR. PLR...etc in pancreatic cancer. However, to our knowledge, this is the first study that delineate the predictive value of these markers for the presence of distant metastases in pancreatic cancer. Other reports focused mainly on the overall survival, progression free survival, response to chemotherapy, the surgical outcomes, or the clinical stage. Also, to our knowledge, this is the first study reporting the prognostic impact of NLR as categorical variable instead of dichotomous variable. Furthermore, to our knowledge, this is the first study that assess the predictive value of NLR for the presence of distant metastases in pancreatic cancer using multiple logistic regression model. 

2. The authors did not comment their results in light to the published data, for example: the identified cutoff of NLR (3.3) is quite different compared to data from literature. Please comment this discrepancy.

Response: Done. We have discussed the differences between NLR cutoff values among the articles, including ours, in the discussion. 

3. More concerns are about patient data. As reported the patient cohort is based on 355 patients. Considering the percentage of absent metastases at baseline from Table 2, the reader can conclude that this experimental group is approximately composed by 178 patients (i.e 89 patients represent 50% of the group). However, from Table 1, patients with baseline metastases (Liver+ lung + peritoneal) are listed as 260. Therefore, the analyzed patient cohort result to be composed by 438 patients. Please comment and clarify this aspect.

Response: Done. We added a sentence to the results and a row to the table #2 showing that 58% of the patients had distant metastases at time of presentation. In our cohort, 206 patients had distant metastases at time of presentation. We edited the results paragraph and the table #2 to avoid any confusion. Thanks for your comment. 

4. Minor concern. Some conclusions are overstated:

Abstract. “…these immune phenomena can help in identifying patients who might respond to immunotherapy”. In my knowledge there are not clinical conventional immune-based approaches for PDAC treatment.

Response: We have edited the manuscript and removed all the sentences that talked about the use of immunotherapy in pancreatic cancer.

5. The graphic quality is very poor.

Response: We used the original graphs in the revised manuscript hoping they are clear now.

---

## [Decision Letter · Decision Letter 1]

7 Apr 2020

The Clinical Value of Peripheral Immune Cell Counts in Pancreatic Cancer

PONE-D-20-00008R1

Dear Dr. Al-Hussaini,

We are pleased to inform you that your manuscript has been judged scientifically suitable for publication and will be formally accepted for publication once it complies with all outstanding technical requirements.

With kind regards,

Aldo Scarpa

Academic Editor

PLOS ONE

Additional Editor Comments (optional):

Reviewers' comments:

Reviewer's Responses to Questions

**Comments to the Author**

1. If the authors have adequately addressed your comments raised in a previous round of review and you feel that this manuscript is now acceptable for publication, you may indicate that here to bypass the “Comments to the Author” section, enter your conflict of interest statement in the “Confidential to Editor” section, and submit your "Accept" recommendation.

Reviewer #1: All comments have been addressed

Reviewer #2: All comments have been addressed

2. Is the manuscript technically sound, and do the data support the conclusions?

Reviewer #1: Yes

Reviewer #2: Yes

3. Has the statistical analysis been performed appropriately and rigorously? 

Reviewer #1: Yes

Reviewer #2: Yes

4. Have the authors made all data underlying the findings in their manuscript fully available?

Reviewer #1: Yes

Reviewer #2: Yes

5. Is the manuscript presented in an intelligible fashion and written in standard English?

Reviewer #1: Yes

Reviewer #2: Yes

6. Review Comments to the Author

Reviewer #1: The authors have well-addressed the issues raised during the revision process in a very complete modality.

Reviewer #2: The authors have satisfied all requests. The new version of the manuscript is now fully satisfactory.I think that this manuscript can be accepted enthusiastically by PLOS ONE readers.

7. PLOS authors have the option to publish the peer review history of their article (what does this mean?). If published, this will include your full peer review and any attached files.

Reviewer #1: No

Reviewer #2: No

---

## [Editor Report · Acceptance letter]

27 May 2020

PONE-D-20-00008R1 

The Clinical Value of Peripheral Immune Cell Counts in Pancreatic Cancer 

Dear Dr. Al-Hussaini:

I am pleased to inform you that your manuscript has been deemed suitable for publication in PLOS ONE. Congratulations! Your manuscript is now with our production department. 

With kind regards,

on behalf of

Dr. Aldo Scarpa 

Academic Editor

PLOS ONE